# The Influence of the Normal Mammary Microenvironment on Breast Cancer Cells

**DOI:** 10.3390/cancers15030576

**Published:** 2023-01-18

**Authors:** Caroline J. Campbell, Brian W. Booth

**Affiliations:** Department of Bioengineering, Clemson University, 401-1 Rhodes Engineering Research Center, Clemson, SC 29634, USA

**Keywords:** breast cancer, microenvironment, redirection, stem cells

## Abstract

**Simple Summary:**

The tumor microenvironment is accepted as a significant part of the tumor progression in many cancers, specifically breast cancer. The complexity of the breast cancer microenvironment is responsible for cancer patient’s response to therapies and, therefore, is the subject of many research studies in breast cancer. The mammary microenvironment is known to transform cells to assume a normal mammary epithelial phenotype. This occurrence is also shown in cancer cells. In a phenomenon called “cancer cell redirection”, tumorigenic cells lose their tumor-forming capacity and differentiate between assuming a normal, non-tumorigenic phenotype. This review will compile the present knowledge of cancer cell redirection and the significant role the normal mammary microenvironment plays on breast cancer cells.

**Abstract:**

The tumor microenvironment is recognized as performing a critical role in tumor initiation, progression, and metastasis of many cancers, including breast cancer. The breast cancer microenvironment is a complex mixture of cells consisting of tumor cells, immune cells, fibroblasts, and vascular cells, as well as noncellular components, such as extracellular matrix and soluble products. The interactions between the tumor cells and the tumor microenvironment modulate tumor behavior and affect the responses of cancer patients to therapies. The interactions between tumor cells and the surrounding environment can include direct cell-to-cell contact or through intercellular signals over short and long distances. The intricate functions of the tumor microenvironment in breast cancer have led to increased research into the tumor microenvironment as a possible therapeutic target of breast cancer. Though expanded research has shown the clear importance of the tumor microenvironment, there is little focus on how normal mammary epithelial cells can affect breast cancer cells. Previous studies have shown the normal breast microenvironment can manipulate non-mammary stem cells and tumor-derived cancer stem cells to participate in normal mammary gland development. The tumorigenic cells lose their tumor-forming capacity and are “redirected” to divide into “normal”, non-tumorigenic cells. This cellular behavior is “cancer cell redirection”. This review will summarize the current literature on cancer cell redirection and the normal mammary microenvironment’s influence on breast cancer cells.

## 1. Introduction

### 1.1. Breast Cancer

Breast cancer research has advanced the understanding of breast cancer development and progression greatly over the past few decades. Despite progress in research, breast cancer is the second leading cause of cancer death in North America and is the most frequent type of cancer for women [1]. Around 40,000 deaths in the United States are due to breast cancer annually [1]. Better treatments and diagnostic tools have increased breast cancer survival rates, but current treatments mainly rely on cytotoxic agents, which decrease a patient’s quality of life due to harsh side effects and sometimes have limited long-term success. 

Breast cancers are categorized into several groups based on their gene expression profile: luminal A, luminal B, basal-like, normal breast-like groups, and breast cancer associated with the human epidermal growth factor receptor (HER)-2 overexpression [2]. The current treatments for breast cancer include chemotherapy, radiotherapy, hormone therapy, targeted therapy, immunotherapy, and mastectomy [3]. This review focuses on HER2-positive breast cancer and triple-negative breast cancer (TNBC). 

### 1.2. HER2-Positive Breast Cancer

Approximately 20–30% of human breast cancers are classified as HER2^+^ [1,2]. Overexpression of HER2^+^ is associated with poor patient outcomes [2]. HER2 is a transmembrane receptor tyrosine kinase encoded by the erbb2 gene located on chromosome 17 and belongs to the epidermal growth factor (EGF) family of receptor tyrosine kinases (RTK). The family of receptors is called erbB in mammals and human EGF receptors (HER) [4]. HER2 activates cell differentiation and proliferation through several transduction pathways [4,5]. HER2 can promote receptor dimerization and increase its own tyrosine kinase activity [4]. HER2 has no known ligand, so dimerization with other HER family members is required for HER2 signaling [4,5]. Because of this ability, HER2 promotes tumorigenesis and is, therefore, classified as an oncogene [4].

In HER2^+^ breast cancer, overexpression of the receptor drives increased cell growth rates, which often leads to a higher probability of metastasis to other tissues. Women with HER2^+^ breast cancer often have rapid disease progression and have poorer outcomes when compared to women whose breast tumors are hormone receptor-positive and HER2^−^ [6,7]. HER2^+^ breast cancer patients often have higher involvement with the lymph nodes and increased resistance to hormone therapy [8].

The most common treatment for HER2^+^ breast cancer is Trastuzumab [9]. Trastuzumab is a monoclonal antibody that binds to the HER2 receptor [9]. This binding causes antitumor effects on HER2+ tissues and the inhibition of HER2 with other HER receptors [9]. Trastuzumab has several antitumor mechanisms. The antibody targets HER2^+^ cancer cells by inducing the downregulation of HER2 receptors and inhibiting HER2-mediated intracellular signaling cascades [9]. Trastuzumab leads to the suppression of HER2^+^ cancer cell growth and proliferation and also recruits effector cells to HER2^+^ tumor sites [9]. Trastuzumab, combined with chemotherapy drugs, greatly improves patient prognosis, along with decreasing patient risk of cancer recurrence and death [9]. Cardiotoxicity and acquired resistance to the therapeutic are challenges for the use of Trastuzumab [9].

### 1.3. Triple-Negative Breast Cancer

Triple-negative breast cancer (TNBC) is an aggressive form of breast cancer, and about 15-20% of all breast cancers are classified as TNBC [2,10]. TNBC is characterized as progesterone receptor-negative (PR), estrogen receptor-negative (ER) and HER2-negative providing the name “triple-negative” to this subtype of cancer [2,10]. About 80% of TNBC breast cancers overlap with the basal-like subtype [2,10]. TNBC is generally more common in African-American women and is associated with the BRCA1 gene mutation [2]. TNBC tends to have a poorer prognosis due to limited treatment options and is more likely to return [2]. The absence of ER, PR, and HER2 expression and identifiable markers renders TNBC difficult to treat due to the lack of specific targets for therapy [10].

The treatment for TNBC comprises surgery, chemotherapy, and radiation [2]. TNBC does not respond to endocrine therapy or HER2-targeted therapies, such as Trastuzumab [2,10]. For advanced stages of TNBC, treatments can include PARP inhibitors, platinum-based chemotherapy drugs, such as carboplatin and cisplatin, and immunotherapy drugs, including atezolizumab and pembrolizumab [10]. For patients with the BRCA mutation, targeted therapies, such as the monotherapy drug Olaparib (PARP inhibitor), or platinum chemo drugs, such as cisplatin, are used when patients no longer respond to common breast cancer chemo drugs [10].

### 1.4. Mammary Gland Development

Mammary gland formation progresses through several phases of development, starting during the embryonic stage and then through major changes during the postnatal stage that are predominantly during puberty. The mammary gland consists of a branching ductal tree in a mammary fat pad [11,12]. The two main cell types occupy the mammary epithelium: basal and luminal [11,12]. The basal epithelium consists of myoepithelial cells and a small population of stem cells. The luminal epithelium forms ducts and secretory alveoli. Accelerated cell turnover occurs during pregnancy, lactation, and involution with rapid differentiation, replication, and apoptosis [11,12]. Constant remodeling of the mammary gland is found throughout one’s lifetime, indicating active stem cells [11,12,13,14]. The mammary gland is often studied for stem cells, microenvironments, and development since the majority of growth and differentiation occurs postnatal and also due to the regular cellular remodeling that occurs during monthly menstrual cycles [12,14].

### 1.5. Mammary Gland Stem Cell Niche

A stem cell niche comprises the microenvironment that encircles stem cells and the immediate cellular activity through short and long distances [14,15]. The mammary microenvironment contains several important aspects, including stem cells, neighboring signaling cells, supporting stroma, extracellular matrix (ECM), and intercellular signals that regulate stem and signaling cells [5,6]. Various cell types comprise the normal mammary niche, including epithelial cells, myoepithelial cells, adipocytes, nerve cells, endothelial cells, and fibroblasts [14,15,16]. Immune cells, such as macrophages, are common as well [14]. Each of these cell types is important for the proper development and maintenance of the mammary gland. For example, in the absence of fatty tissue, branching morphogenesis is severely restricted, which is the key to proper ductal tree development [17,18].

Stem cell differentiation is controlled by heterologous cell–cell interactions from the surrounding cells through numerous biochemical and biophysical factors [14]. The microenvironment directly influences proper development through hormones, growth factors, non-multipotent cells, and ECM composition. Many secreted factors control the indirect communication of stem cells and surrounding niche cells. Hormones and growth factors control normal mammary gland development. Ductal morphogenesis is controlled by estrogen, and members of the EGF family of growth factors controls. Progesterone controls the branching of ducts as well as a driving force for progression through the mammary stem cell hierarchy. [11,12]. The Wnt3A and Wnt4 cytokines monitor progesterone production [19,20]. Prolactin and the erbB4 receptor are involved in proper milk production [12,14]. ECM composition of the mammary gland niche is also important for directing stem cells, particularly through integrins [14,18,21]. Integrins regulate pathways between the ECM receptors, stroma, and surrounding cells and can directly influence mammary gland development [18]. The mammary microenvironment affects proper mammary gland development and proliferation but is also known to drive tumor progression [22]. The tumor microenvironment drives tumor progression and is possibly a determining factor in tumor response to chemotherapeutic agents.

## 2. Cell Redirection

### 2.1. Cell Redirection by Mammary Microenvironments In Vivo

Each cell type in the mammary microenvironment influences cell growth, homeostasis, and normal development through a range of intercellular signals. The intracellular signals mediate cell differentiation and conceivably prevent tumor formation through apoptotic and anti-proliferative signals to control irregular cell growth. Disruption of these signals or pathways can lead to uncontrolled cell proliferation and tumor formation.

The mammary microenvironment influences non-mammary stem cells, isolated stem cells from the central nervous system (CNS), bone marrow (BM), testes, or embryonic stem cells (ESCs). When non-mammary stem cells are co-transplanted with normal mammary epithelial cells (MECs) the non-mammary cells adopted a normal mammary phenotype in the in vivo mouse model of mammary gland regeneration (Figure 1) [23,24,25,26,27]. This phenomenon was named “cellular redirection”. The non-mammary stem cells participate in the formation of mammary stem cell niches and divide into differentiating mammary epitheliums that include myoepithelial cells and milk-producing secretory cells [23,24,25,26,27]. The non-mammary somatic stem cells respond to intercellular signals from the normal mammary microenvironment [23,24,25,26,27]. When the non-mammary stem cells are transplanted alone, no normal mammary outgrowths are observed, and the embryonic stem cells form teratomas [23,24,25,26,27]. This indicates that signals arising from mammary stroma alone are not sufficient to direct non-mammary stem cell growth and differentiation into functional mammary cells and tissue. These findings demonstrate that the deterministic nature of the normal mammary microenvironment laid the groundwork for the cancer cell redirection discussed in the following sections.

### 2.2. Cancer Cell Redirection

The equivalent model of mammary gland regeneration has been used in conjunction with tumor-derived cancer stem cells (CSCs) (Figure 2). Breast cancer patients with overexpression of the neu oncogene are associated with higher rates of tumor formation, and the MMTV-neu mouse model is a recognized model for studying HER2^+^ human breast cancer [16]. In this model, transgenic mice that express wild-type neu under the transcriptional regulation of the mouse mammary tumor virus-long terminal repeat promotor (MMTV-LTR) are bred with WAP-CRE/Rosa26R mice [16]. Tumor cells from the WAP-Cre/Rosa26R/MMTV-neu mice repeatedly express LacZ. After injection of tumor cells at different concentrations into cleared mammary fat pads of 3-week-old Nu/Nu mice, LacZ^+^ tumors arose in 100% of cases within 7 months [16]. LacZ^+^ tumor cells were co-transplanted with normal MECs at specific ratios of 2:1, 1:5, and 1:50 ratios, respectively [28]. LacZ^+^ mammary tumors arose in all co-transplanted ratios except 1:50. No tumor development was noticed in 1:50 transplants with 1000 MMTV-neu cells and 50,000 MECs even though outgrowths contained LacZ^+^ tumor-derived cells in luminal and basal locations. The LacZ^+^ cells differentiated to express ER, PR, or smooth muscle actin. Additionally, the tumor-derived LacZ^+^ cells produced milk proteins during pregnancy and lactation. These remained following mammary involution and, in second-generation transplantation, again contributed to normal mammary structures. This outcome indicates that, in this condition, signals produced by normal mammary microenvironments are capable of suppressing the tumorigenic phenotype of WAP-Cre/Rosa26R/MMTV-neu tumor-derived cells [29]. This phenomenon is named “cancer cell redirection”. The tumorigenic cells are redirected to form normal mammary structures and lose their ability to form tumors [14].

The same animal model for cancer cell redirection was used in order to establish cancer cell redirection using human cancer cells. TNBC cells (MDA-MB-231 and MBA-MB-468) were transplanted alone, in a 1:1 ratio with MECs, or a 1:50 ratio with MECs [30]. The TNBC cells transplanted were CD44^+^/CD24^−^, indicating they were prospective breast cancer stem cells. Mammary tumors formed when TNBC cells were transplanted alone or in a 1:1 ratio with MECs, but no tumors formed when transplanted in a ratio of 1:50 with MECs. The TNBC cells transplanted in a 1:50 ratio with MECS were redirected to form normal mammary ductal trees [30]. The redirected TNBC lost their tumor-forming capacity and differentiated to express normal breast epithelial markers and produced milk proteins during pregnancy and lactation.

The normal mouse mammary microenvironment also redirected CSCs derived from human embryonal testicular carcinoma, NTERA-1 cl (NT2) [31]. Tumors formed when NT2 cells were transplanted alone or transplanted in ratios of 1:1 with MECs. When NT2 cells were transplanted with MECs in ratios of 1:10 and 1:50, NT2 cells differentiated into luminal, basal (myoepithelial), and secretory cells [31]. These outcomes suggest that totipotent human embryonal carcinoma cells are redirected in the mouse mammary gland to adopt a normal human mammary epithelial phenotype in the absence of tumorigenic activity [31].

These results suggest that cancer cell redirection induced by a normal mammary microenvironment is not limited to cells of mouse origin or cells of mammary origin [16,31,32]. However, the ratio of normal cells (MECS) to cancer cells will control cancer cell redirection in transplantations. A summary of these findings can be seen in Table 1 [15,16,30,31].

### 2.3. Differential Gene Expression in Mouse Mammary Microenvironment In Vitro

Through asymmetric division, stem cells self-renew by maintaining their template DNA strands while passing the newly synthesized DNA to the daughter cells [32]. Self-retaining of the template strands is how stem cells are hypothesized to protect themselves from DNA replication mutations and potential cancer risks. Park et al. used asymmetric division of stem cells to identify signaling pathways differentially expressed in self-renewing mouse mammary stem cells [32]. Newly forming mouse mammary stem cells were labeled with the thymidine analog 5-ethynl-2′-deoxyuridine during pubertal mammary ductal expansion [32]. Label-retaining cells (LRCs) were defined as cells that maintained the DNA nuclear label after extended chase periods [32]. After euthanasia, mammary cells were collected and sorted based on the nuclear label [32]. Notch and Wnt signaling pathways were differently expressed compared to non-LRCs. Hes1 and Hey2, Notch-inducible genes, were elevated in LRCs [32]. Reduced colony formation and reduced label retention of mammary epithelial cells in vitro were seen with the inhibition of Notch1 by shRNA. [32]. Notch and Wnt signaling pathways are involved in the regulation of stem cells in the mouse mammary gland. Though the Notch pathway is important for normal mammary development, the Notch pathway also plays a role in breast cancer development. Overexpression of the Notch receptors correlates with cancer initiation and progression. Results from Park et al. suggest that genes in the LRCs of the mammary gland are differently regulated than non-LRCs, and Notch1 moderates asymmetric cell division in mammary progenitor cells.

### 2.4. Cell Redirection by Mammary Microenvironment In Vitro

After implantation and redirection of WAP-Cre/Rosa26R/MMTV-neu tumor-derived cells in vivo from the 1:50 ratio subgroup, erbB2 continued to be overexpressed, but the phosphorylation of erbB2 was absent, indicating altered intracellular signal transduction pathways [16]. ErbB2 was phosphorylated in implanted cancer cells where tumors formed but was absent in the redirected cells [16]. The attenuation of erbB2 phosphorylation serves as a biomarker of redirection though not as a mechanism of cancer cell redirection. This biomarker of attenuation of erbB2 phosphorylation in redirected cells was used to establish cancer cell redirection for in vitro models [33].

An in vitro model was established to mimic the in vivo mammary microenvironment in order to test cancer cell redirection [33]. The in vitro model was validated using a normal mouse mammary epithelial cell line, COMMA-Dβgeo, and the tumor cell lines derived from MMTV-neu [33]. MMTV-neu cells expressed erbB2 and p-erbB2 when grown alone, while COMMA-Dβgeo expressed very little erbB2 and no p-erbB2 when cultured alone [33]. When MMTV-neu cells were co-cultured with COMMA-Dβgeo cells in a ratio of 1:50, the expression of p-erbB2, but not erbB2, was diminished in MMTV-neu cells [33]. These findings are consistent with previous in vivo mouse model studies where the attenuation of p-erbB2 led to redirected tumor-derived cells and their incorporation into mammary outgrowth and no tumor formation [16].

The in vitro model assessed the redirection capacity of human breast cancer cells using human breast epithelial cells (MCF10A and MCF12 cells) and human HER2^+^ breast cancer cells (SkBr3, BT474, HCC1954). HER2+ breast cancer cells and epithelial cells were cultured either alone or in ratios of 1:1 or 1:50, respectively. When cultured alone, HER2^+^ breast cancer cells expressed both HER2 and phospho-HER2 [15]. The breast epithelial cells do not express noticeable levels of HER2 or phospho-HER2 in vitro [15]. The HER2^+^ breast cancer cells continue to express both HER2 and phospho-HER2 when the two cell types are co-cultured in a 1:1 ratio [15]. When the two cell types are co-cultured using the redirection ratio of 1:50, the HER2^+^ breast cancer cells express HER2, but phosphorylation of the HER2 receptor is absent [15]. The results imply HER2^+^ breast cancer cells have undergone phenotype redirection. To establish if apoptosis was a contributing factor to the results, HER2^+^ breast cancer cells were treated with doxorubicin and induced apoptosis in the HER2^+^ breast cancer cells [15]. The untreated cancer cells and redirected cancer cells had low levels of apoptosis when compared to the treated cells, suggesting that apoptosis is not a major factor in cancer cell redirection in vitro [15].

### 2.5. Phenotypic Changes Induced through In Vitro Redirection

Phenotypic changes were observed in HER2^+^ breast cancer cells after redirection in both in vivo and in vitro models. It was not known if the phenotype change was permanent. To test for permanent phenotype changes, monocultures and co-cultures of MECs and HER^+^ breast cancer cells in ratios of 1:1 and 1:50 were grown [15]. The HER2^+^ breast cancer cells were transduced to constitutively express red fluorescent protein (RFP), allowing for tracking of the cancer cells in vivo. Using HER2 expression, the various cell types were magnetically sorted, and the sorted fractions were then transplanted in cleared mammary fat pads of 3-week-old athymic female mice. The transplantation of HER2^+^ cancer cells resulted in mammary tumor formation, while the transplantation of normal MECs resulted in normal mammary development. Tumor formation was found in all animals that received HER2^+^ cells sorted from 1:1 co-cultures though tumor formation was delayed compared to cancer cells alone [15]. Normal epithelial growth derived from RFP+ cells was also found in 75% of the animals that received HER2^+^ cancer cells from 1:1 co-cultures [15]. No tumor formation occurred, and normal RFP^+^ epithelial growth was found in animals that received the sorted HER2^+^ RFP^+^ cells from 1:50 co-cultures [15]. These results indicate that phenotypic change is maintained after the redirection of HER2^+^ cancer cells and transplantation [15].

### 2.6. Gene Expression Profile Changes In Vitro Redirection

Though phenotypic changes are established after cancer cell redirection, a comprehensive gene expression profile is not well-known. Gene expression was explored using the in vitro redirection model and the known overexpression of HER2 in HER2^+^ breast cancer cells that undergo redirection [15,16,32]. Co-cultures of 1:1, 1:50, and monocultures of HER2^+^ breast cancer cells and breast epithelial cells were sorted based on HER2 expression. The sorted fractions were applied to RNAseq analysis. Data analysis was performed using R/Bioconductor software package *limma* following RNA sequencing in order to read, normalize the data set, and implement differential expression analyses. The gene expression profiles showed patterns particular to both cancer cells and epithelial control cells.

The data revealed that more than half of the genes from the RNAseq analysis are significantly differentially expressed (DE) between the HER2^+^ breast cancer and epithelial cell lines [15]. The gene for CD44 was a notable DE gene from the analysis. CD44, which is often used as a biomarker for breast cancer stem cells, was significantly differentially expressed between the cancer cells and epithelial cells and between the cancer cells and redirected cells [15]. CD44 is part of several pathways that were significantly different between the cancer cells and epithelial cells and between the cancer cells and redirected cells. There was no significant difference in CD44 gene expression between the epithelial cells and the redirected cells. Some of the differentially expressed pathways are the EMT, TNFα_via_NF_K_B, apoptosis, IL6_JAK_STAT3, and IL2_STAT5 pathways [15]. Similar results were found when pathway analysis was performed on redirected MMTV-neu mouse mammary tumor cells [34]. In both in vitro redirection and in vivo redirection of Notch1, Notch2, Wnts, and Hedgehog family members are differentially expressed between normal epithelial cells and redirected cells when compared to cancer cells (Figure 3) [15,34,35]. Multiple receptors, including EGFR and erbB3, as well as TGFβ1 and TGFβ2, are downregulated in redirected cells when compared to the cancer cells. Another factor involved is ECM from normal epithelial cells [36].

The HER2^+^ redirected cells had a similar gene profile to the mammary epithelial cells and a significantly different profile from the HER2^+^ breast cancer cell line [15]. These results indicate epithelial cells are supplying signals that affect HER2^+^ breast cancer cells, and the redirected cells adopt a gene expression profile similar to a normal epithelial profile, including changes in intracellular pathways. These modifications in intracellular signaling pathways could lead to a better understanding of cancer mechanisms and, therefore, better treatment options for breast cancer patients.

## 3. Conclusions

The intercellular signals and pathways from a normal mammary microenvironment stimulate and support the proper development and maintenance of the mammary gland. However, disruption of these intercellular signals and pathways could cause uncontrolled cell proliferation, tumor formation, and cancer. The influence of the normal mammary microenvironment is shown by the transformation of non-mammary stem cells into mammary outgrowths after transplantation into normal mammary tissue. Mammary microenvironments influence cancer cells to undergo a phenotypic shift and lose their tumor-forming ability in both in vitro and in vivo models. The phenomenon is called “cancer cell redirection”. With these characteristics established, intercellular signals from normal mammary epithelium induce phenotypic changes in mammary and non-mammary stem cells, including cancer stem cells. However, the system used to study cancer cell redirection is limited due to its artificial nature, and there is no evidence that redirection occurs in breast cancer patients. Despite limitations, these findings have the potential to identify the epithelial-derived signals that suppress cancer cell growth and induce differentiation into normal epithelial phenotype, and, therefore, improve the understanding of the function of the mammary microenvironment.

## 4. Future Directions

This review summarizes that cancer cell redirection can be achieved in both in vitro and in vivo systems, but the exact mechanisms are still unknown. Narrowing down the soluble factors or cellular signals that influence this process will further the understanding of the breast cancer microenvironment’s impact on cancer progression. Future studies should focus on the individual signals and pathways that potentially cause this phenotypic transformation in cancer cell redirection. Once identified, these signals or pathways can be potential therapeutic targets for breast cancer treatment or targets in combination with current cancer treatments to help reduce cancer progression or even eliminate cancer cells.

## Figures and Tables

**Figure 1 cancers-15-00576-f001:**
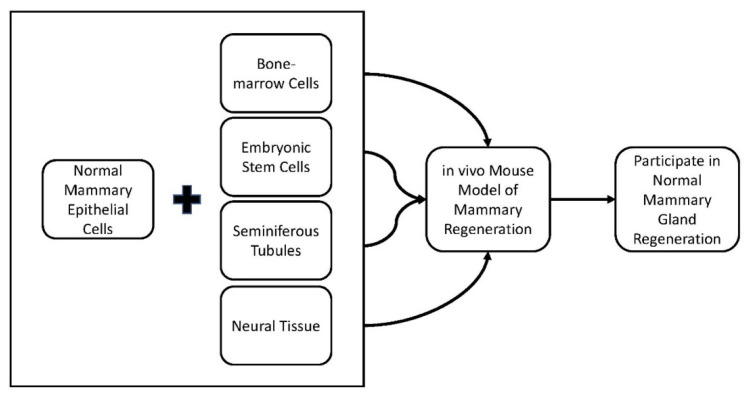
Outline of how multiple studies have demonstrated that the normal mammary microenvironment can redirect non-mammary stem cells when placed in the in vivo mouse model of regeneration to participate in normal mammary gland regeneration [3,23,24,25,26,27].

**Figure 2 cancers-15-00576-f002:**
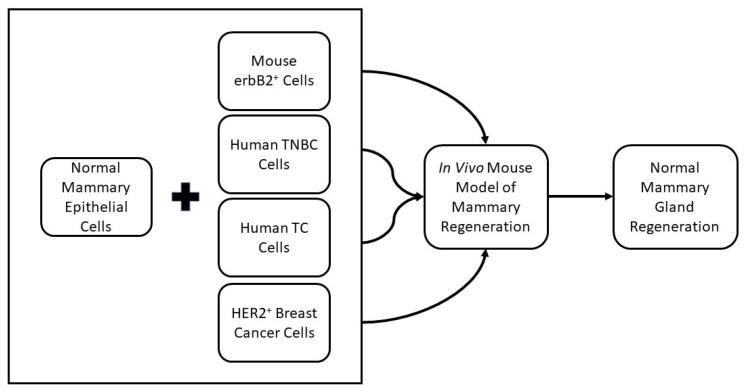
Outline of how multiple studies have demonstrated that the normal mammary microenvironment can redirect cancer cells when placed in the in vivo mouse model of regeneration to participate in normal mammary gland regeneration [15,16,30,31]. TC—Testicular Carcinoma, TNBC—Triple-Negative Breast Cancer.

**Figure 3 cancers-15-00576-f003:**
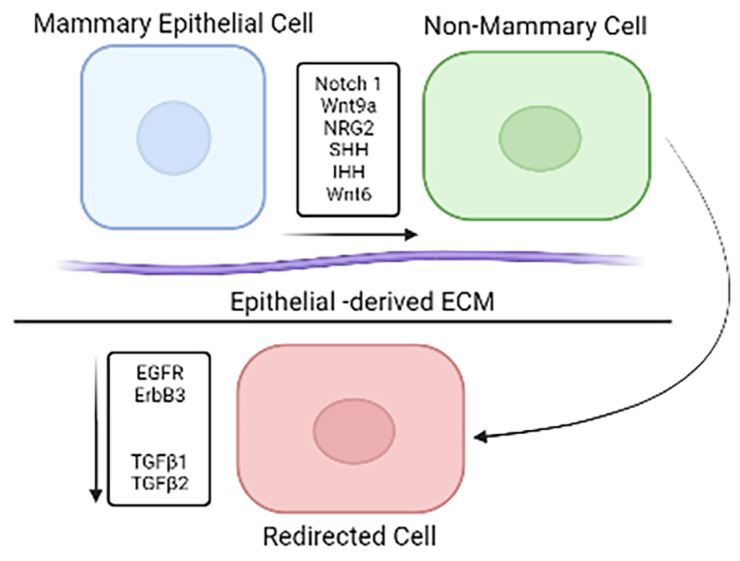
Mammary epithelial cells induce gene expression changes.

**Table 1 cancers-15-00576-t001:** Compilation of results of studies where cancer cells are redirected by the normal mammary microenvironment [15,16,30,31].

Cells Transplanted	Ratio of Cancer Cells:MECs	# Positive Takes/# Implants	# Cancer Cell Takes/# Positive Takes	% Tumors
Mouse erbB2^+^ Cancer Cells Only	Cancer only	0/18	0/0	100
Mouse erbB2^+^ Cancer Cells with MECs	1:50	15/16	15/15	6.25
Human TC Cells Only	Cancer only	0/6	0/0	33.3
Human TC Cells with MECs	1:50	10/12	10/10	0
Human TNBC Cells with MECs	1:5	5/20	5/5	50
Human TNBC Cells with MECs	1:50	10/16	10/10	0
HER2^+^ Breast Cancer Cells Only	Cancer only	0/4	0/0	100
HER2^+^ Breast Cancer Cells with MECs	1:50	2/4	2/4	0

TC—Testicular Carcinoma, TNBC—Triple-Negative Breast Cancer.

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
