# Peer review of "The Influence of the Normal Mammary Microenvironment on Breast Cancer Cells"

_cancers, 2023, doi:10.3390/cancers15030576_

Round 1

Reviewer 1 Report

I recommend adding a couple of sections on limitations of the reviewed studies as well as current and future potential applications of cancer cell redirection.

Author Response

Reviewer 1

I recommend adding a couple of sections on limitations of the reviewed studies as well as current and future potential applications of cancer cell redirection.

RESPONSE: We added more information on future potential applications of cancer cell redirection in a new section at the end of the manuscript. Limitations have been added to the Conclusion.

Reviewer 2 Report

In this manuscript, the authors have described the role of normal mammary microenvironment on breast cancer cells, especially for HER2+ breast cancer cells. Overall, the authors have fully described the “cancer cell redirection” phenomenon. However, there are some questions still need to be answered:

1) The authors only described the research in HER2+ subtype breast cancer. Does it only exist in this subtype or a universal phenomenon in all subtypes of breast cancer? The authors mainly discussed the studies in HER2+ and TNBC.

2) Could the authors include more information about other subtypes of breast cancer? For example, TNBC in Introduction part?

Author Response

Reviewer 2

In this manuscript, the authors have described the role of normal mammary microenvironment on breast cancer cells, especially for HER2+ breast cancer cells. Overall, the authors have fully described the “cancer cell redirection” phenomenon. However, there are some questions still need to be answered:

1) The authors only described the research in HER2+ subtype breast cancer. Does it only exist in this subtype or a universal phenomenon in all subtypes of breast cancer? The authors mainly discussed the studies in HER2+ and TNBC.

RESPONSE: On pages 4-5 we outlined the research using TNBC and testicular carcinoma cells.

2) Could the authors include more information about other subtypes of breast cancer? For example, TNBC in Introduction part?

RESPONSE: We feel the review should focus on cellular redirection and not breast cancer subtypes.

Reviewer 3 Report

1.)    Page 1. A Materials and Methods section should be included to indicate how these papers were selected for review.

2.)    Page 3, line 99-100. Progesterone controls the branching of ducts. While this statement is correct, it is probably worth mentioning that progesterone from the menstrual cycle, pregnancy, exogenous sources and others is a major driving force for progression through the mammary stem cell hierarchy, acting indirectly in a paracrine manner through mammary epithelial, as well as directly on multipotent progenitor cells, luminal progenitor cells, and others.  

3.)    Page 3, line 94ff. This paragraph is also very diverse, and might be more organized if the purpose is to describe control of stem cell differentiation.

4.)    Page 4, line 134. This statement indicates that the model of transplanting normal mammary epithelial cells and cancer stem cells into an vivo mouse model has been studied. Are these CD44+/CD24- breast cancer stem cells? Please provide the data for these studies.

5.)    The paragraph 134-150 is difficult to follow: It is stated that when LacZ+ cells and HMEC are cultured in a ratio of 1:50, tumor growth is suppressed, apparently due to "signals", although these signals are not described. It goes on to say that the tumorigenic cells are "redirected" to form normal mammary structures, although no data is provided to support this statement. Please provide more details regarding these structures. Do these tumorigenic cells contain cancer stem cells?

6.)    Page 4, line 156. What are the histological characteristics of the redirected TNBC cells? Have all malignant characteristics been lost, including histologic?

7.)    Page 7, line 219ff. It is difficult to relate these studies, where normal mammary epithelial cells redirect breast cancer cell lines which are transplanted into mice, to events that may be happening in a breast cancer in women. One could argue that this is an artificial system. Breast cancers arise within a breast parenchyma of cancerized fields, where the normal breast tissue contains many genomic changes and is at high risk for breast cancer because of these changes. The associated high risk epithelial cells would appear to have little or no resemblance to the HMECs, such as MCF10A which are developed from normal breast tissue and are used in these experiments. What is the evidence that Redirection of tumor cells is taking place within a breast cancer in women?

8.)    It would be helpful to describe how these findings, especially the “redirection” of tumor cells, could be studied in breast cancer.

9.)    Page 8, Conclusion: It is concluded that "Mammary microenvironments influence cancer cells to undergo a phenotypic shift....." In this study the mammary microenvironment is that of a cleared mammary fat pad in mice, populated by normal HMECs and cells from breast cancer cell lines. This “microenvironment” in some manner causes the malignant cells to “redirect. Unfortunately, no evidence is presented that similar changes have been observed within a breast cancer. Further, the tumor microenvironment in breast cancer, with it's tumor infiltrating lymphocytes, cancer-associated fibroblasts, cancer-associated adipocytes, cancer-associated macrophages and a variety of other cellular and systemic changes is clearly very different from the microenvironment consisting of HMECs and breast cancer cell lines in a cleared mammary fat pad. These differences should be discussed, and why the authors feel these Redirection" changes are likely to occur in cancer.

Author Response

Reviewer 3

Page 1. A Materials and Methods section should be included to indicate how these papers were selected for review.

RESPONSE: We added a materials and methods section after the Introduction. 

2.)    Page 3, line 99-100. Progesterone controls the branching of ducts. While this statement is correct, it is probably worth mentioning that progesterone from the menstrual cycle, pregnancy, exogenous sources and others is a major driving force for progression through the mammary stem cell hierarchy, acting indirectly in a paracrine manner through mammary epithelial, as well as directly on multipotent progenitor cells, luminal progenitor cells, and others.  

RESPONSE: We added this information to the article.

3.)    Page 3, line 94ff. This paragraph is also very diverse, and might be more organized if the purpose is to describe control of stem cell differentiation.

RESPONSE: This paragraph has been edited to attempt to alleviate confusion.

4.)    Page 4, line 134. This statement indicates that the model of transplanting normal mammary epithelial cells and cancer stem cells into an vivo mouse model has been studied. Are these CD44+/CD24- breast cancer stem cells? Please provide the data for these studies.

RESPONSE: Transplanted TNBC were CD44+/CD24- breast cancer stem cells. Data from these studies has been added as suggested.

5.)    The paragraph 134-150 is difficult to follow: It is stated that when LacZ+ cells and HMEC are cultured in a ratio of 1:50, tumor growth is suppressed, apparently due to "signals", although these signals are not described. It goes on to say that the tumorigenic cells are "redirected" to form normal mammary structures, although no data is provided to support this statement. Please provide more details regarding these structures. Do these tumorigenic cells contain cancer stem cells?

RESPONSE: Additional information describing the normal mammary structures has been added. The signals that direct cancer cell redirection are currently unknown.

6.)    Page 4, line 156. What are the histological characteristics of the redirected TNBC cells? Have all malignant characteristics been lost, including histologic?

RESPONSE: This information has been added as requested.

7.)    Page 7, line 219ff. It is difficult to relate these studies, where normal mammary epithelial cells redirect breast cancer cell lines which are transplanted into mice, to events that may be happening in a breast cancer in women. One could argue that this is an artificial system. Breast cancers arise within a breast parenchyma of cancerized fields, where the normal breast tissue contains many genomic changes and is at high risk for breast cancer because of these changes. The associated high risk epithelial cells would appear to have little or no resemblance to the HMECs, such as MCF10A which are developed from normal breast tissue and are used in these experiments. What is the evidence that Redirection of tumor cells is taking place within a breast cancer in women?

RESPONSE: Currently there is no evidence that redirection is occurring in breast cancer patients. The goal of these studies is to identify epithelial-derived signals that suppress cancer cell growth and induce differentiation. Once these are identified they can be targeted, perhaps in conjunction with current treatments, to help slow progression or eradicate tumor cells.

8.)    It would be helpful to describe how these findings, especially the “redirection” of tumor cells, could be studied in breast cancer.

RESPONSE: We added more information on future potential applications of cancer cell redirection.

9.)    Page 8, Conclusion: It is concluded that "Mammary microenvironments influence cancer cells to undergo a phenotypic shift....." In this study the mammary microenvironment is that of a cleared mammary fat pad in mice, populated by normal HMECs and cells from breast cancer cell lines. This “microenvironment” in some manner causes the malignant cells to “redirect. Unfortunately, no evidence is presented that similar changes have been observed within a breast cancer. Further, the tumor microenvironment in breast cancer, with it's tumor infiltrating lymphocytes, cancer-associated fibroblasts, cancer-associated adipocytes, cancer-associated macrophages and a variety of other cellular and systemic changes is clearly very different from the microenvironment consisting of HMECs and breast cancer cell lines in a cleared mammary fat pad. These differences should be discussed, and why the authors feel these Redirection" changes are likely to occur in cancer.

RESPONSE: Please see our answer to comment #7 above.

Reviewer 4 Report

The describe a novel 'paradigm-shift' cell redirection phenomenon in tumorigenesis. In principle this is an amazing concept. I have only three Major concerns:

1) The authors are pioneers and have extensive experience in publishing original research work on this novel phenomenon. However, unfortunately, the way the manuscript is written, is largely description of their previous work. In a review article, the authors should provide some futuristic prospects and possible implications on current therapeutic strategies. Do authors envision a supportive role of cell redirection following immunotherapy (which does not get rid of stem cells)? Do the authors have any mechanistic role of this application in current chemotherapy? Or is there any biomarker/diagnostic implication. I feel the authors are underselling their work by not stressing on the futuristic significance of this phenomenon.

2) For the 2.5 and 2.6 portions of the manuscript, the authors should provide a image/figure to explain the cell-cell interaction, explaining the role of possible cytokines and transcription factors impacting this phenomenon. Even if there is no direct experimental/research evidence, the authors could provide a predictive model to explain how they think this phenomenon is happening? 

3) The cell-tumorigenic process has internal metabolic and external immunologic impact towards tumor differentiation in tumor microenvironment. What is the authors' take on the impact of immunologic/metabolic effect towards normalization remodeling in this 'cell-redirection' phenomenon.  

Author Response

Reviewer 4

The describe a novel 'paradigm-shift' cell redirection phenomenon in tumorigenesis. In principle this is an amazing concept. I have only three Major concerns:

1) The authors are pioneers and have extensive experience in publishing original research work on this novel phenomenon. However, unfortunately, the way the manuscript is written, is largely description of their previous work. In a review article, the authors should provide some futuristic prospects and possible implications on current therapeutic strategies. Do authors envision a supportive role of cell redirection following immunotherapy (which does not get rid of stem cells)? Do the authors have any mechanistic role of this application in current chemotherapy? Or is there any biomarker/diagnostic implication. I feel the authors are underselling their work by not stressing on the futuristic significance of this phenomenon.

RESPONSE:  We added more information on future potential applications of cancer cell redirection

2) For the 2.5 and 2.6 portions of the manuscript, the authors should provide a image/figure to explain the cell-cell interaction, explaining the role of possible cytokines and transcription factors impacting this phenomenon. Even if there is no direct experimental/research evidence, the authors could provide a predictive model to explain how they think this phenomenon is happening? 

RESPONSE: We added a figure and text describing for potential cell to cell interactions. 

3) The cell-tumorigenic process has internal metabolic and external immunologic impact towards tumor differentiation in tumor microenvironment. What is the authors' take on the impact of immunologic/metabolic effect towards normalization remodeling in this 'cell-redirection' phenomenon.  

RESPONSE: That is an aspect we are currently researching.

Round 2

Reviewer 3 Report

The format for replying to the concerns of the Reviewers is inadequate. The specific location in the manuscript where each revision is provided should be indicated in the response, and a detailed description of each change should be provided in the response.

The Materials and Methods section revision is inadequate. This should include the topics and subject headings that are used so that the comprehensiveness of the search can be determined and this search can be reproduced by interested readers.

The reviewer recommended including more information about other subtypes of breast cancer. This has not been provided. Please provide this information.

Page 133-145, effect of mammary microenvironment on non-mammary stem cells. This paragraph is confusing, but more importantly, it is unclear what modulation of non-mammary stem cells by normal microenvironment has to do with Breast cancer. This paragraph could easily be eliminated.

Item #9. The answer to Comment #7 does not answer the concern in Comment #9.  The point the reviewer is making is that the authors are studying a microenvironmental system in vitro/in vivo which is very different from the TME of a breast cancer, but the characteristics of the breast cancer TME are not described, and the differences between the experimental system are not described, discussed, or even acknowledged. This needs to be addressed, especially if the experimental system is to be used to identify targets which may be studied in breast cancer.

Author Response

The Materials and Methods section revision is inadequate. This should include the topics and subject headings that are used so that the comprehensiveness of the search can be determined and this search can be reproduced by interested readers.

RESPONSE: We have looked at over 20 other review articles published in Cancers and in no instance did we find a Materials and Methods section as requested by the reviewer. We have asked the Editor to make a decision on whether this addition is necessary since it has not appeared in review articles published in this journal previsouly. We have removed our initial addition until we receive the Editor’s decision.

The reviewer recommended including more information about other subtypes of breast cancer. This has not been provided. Please provide this information.

RESPONSE: In our response to the reviewer who recommended this addition we stated that the focus of this review is on cellular redirection and not the different subtypes of breast cancer. That satisfied the reviewer. However, breast cancer redirection has been demonstrated in both HER2-positive breast cancer and TNBC (as well as testicular carcinoma) so we have added TNBC background. We have added a new subsection, 1.3, where we added information on triple-negative breast cancer.

Page 133-145, effect of mammary microenvironment on non-mammary stem cells. This paragraph is confusing, but more importantly, it is unclear what modulation of non-mammary stem cells by normal microenvironment has to do with Breast cancer. This paragraph could easily be eliminated.

RESPONSE: This paragraph details the initial findings on which cancer cell redirection is based. We have added additional lines within and at the end of subsection 2.1 to emphasize this point.

Item #9. The answer to Comment #7 does not answer the concern in Comment #9.  The point the reviewer is making is that the authors are studying a microenvironmental system in vitro/in vivo which is very different from the TME of a breast cancer, but the characteristics of the breast cancer TME are not described, and the differences between the experimental system are not described, discussed, or even acknowledged. This needs to be addressed, especially if the experimental system is to be used to identify targets which may be studied in breast cancer.

RESPONSE: Thank you for the clarification of your initial critiques. We agree with the reviewer that the normal microenvironment and the TME are vastly different. In no way are we attempting to compare the two. Conversely, we are describing the effects that normal microenvironments have on normal, non-mammary cells and cancer cells leading changes in phenotype of the cells introduced into the experimental system. Sentences addressing this have been added to the end of the Conclusion to clarify this valid point.